# Uni-Sign: Toward Unified Sign Language Understanding at Scale

**Zecheng Li**[1], **Wengang Zhou**[1,2†], **Weichao Zhao**[1], **Kepeng Wu**[1], **Hezhen Hu**[3], **Houqiang Li**[1,2]

[1] MoE Key Laboratory of Brain-inspired Intelligent Perception and Cognition,
   University of Science and Technology of China

[2] Institute of Artificial Intelligence, Hefei Comprehensive National Science Center

[3] University of Texas at Austin

{lizecheng23, saruka, wukp}@mail.ustc.edu.cn
{zhwg, lihq}@ustc.edu.cn, alexhu@utexas.edu

## Abstract

Sign language pre-training has gained increasing attention for its ability to enhance performance across various sign language understanding (SLU) tasks. However, existing methods often suffer from a gap between pre-training and fine-tuning, leading to suboptimal results. To address this, we propose Uni-Sign, a unified pre-training framework that eliminates the gap between pre-training and downstream SLU tasks through a large-scale generative pre-training strategy and a novel fine-tuning paradigm. First, we introduce CSL-News, a large-scale Chinese Sign Language (CSL) dataset containing 1,985 hours of video paired with textual annotations, which enables effective large-scale pre-training. Second, Uni-Sign unifies SLU tasks by treating downstream tasks as a single sign language translation (SLT) task during fine-tuning, ensuring seamless knowledge transfer between pre-training and fine-tuning. Furthermore, we incorporate a prior-guided fusion (PGF) module and a score-aware sampling strategy to efficiently fuse pose and RGB information, addressing keypoint inaccuracies and improving computational efficiency. Extensive experiments across multiple SLU benchmarks demonstrate that Uni-Sign achieves state-of-the-art performance across multiple downstream SLU tasks. Dataset and code are available at github.com/ZechengLi19/Uni-Sign.

## 1 Introduction

Sign languages are the primary means of communication for the Deaf/Hard of Hearing individuals, conveyed via hand gestures, facial expressions, and movements (Braem & Sutton-Spence, 2001). Considering the critical benefits for barrier-free communication between Deaf/Hard of Hearing and hearing communities, sign language understanding (SLU) has been extensively studied for decades (Camgoz et al., 2018; Yin et al., 2021). SLU presents unique challenges that necessitate a comprehensive understanding of the visual cues embedded in individual sign signals, as well as the distinctive linguistic rules of sign language. SLU can be subdivided into several sub-tasks, including isolated sign language recognition (ISLR), continuous sign language recognition (CSLR), and sign language translation (SLT). ISLR concentrates on classifying individual sign language movements, while CSLR aims to learn the alignment of sequences between sign language and their corresponding glosses. In contrast, SLT requires the model to generate textual descriptions corresponding to sign language sequences. All these tasks impose indispensable demands on the model's fine-grained comprehension and context awareness capabilities.

Recently, more and more studies have shifted their attention towards the exploration of pre-training techniques for SLU, which benefit from large-scale data to learn discriminative representations. One main thread attempts to utilize large-scale self-supervised learning to unleash the statistics in unlabeled data (Hu et al., 2021a; 2023a; Zhao et al., 2024b). SignBERT+ designs self-supervised learning strategies in a masking-and-reconstructing manner. Although these methods demonstrate

---

† Corresponding author.

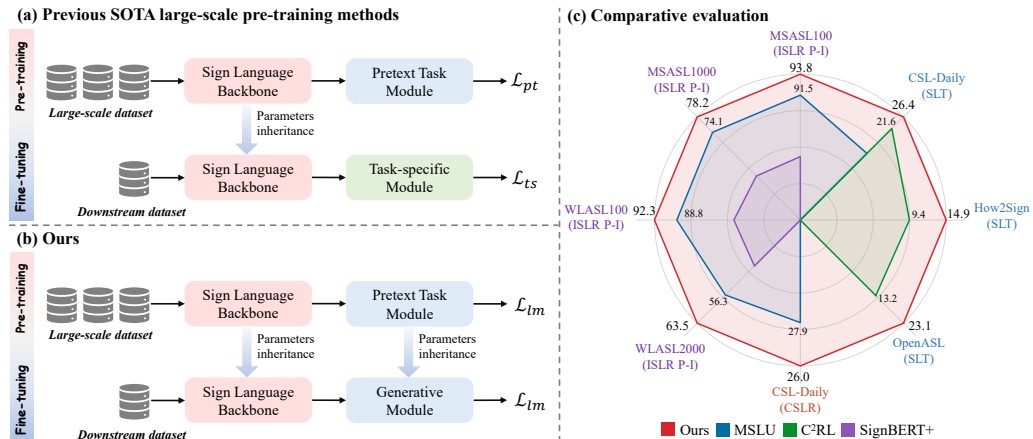

Figure 1: Comparison of paradigm and performance between previous SOTA pre-training methods and ours. $\mathcal{L}_{pt}$, $\mathcal{L}_{ts}$, and $\mathcal{L}_{lm}$ represent the pretext-task loss, task-specific loss, and language modeling loss, respectively. Our method could mainly adopt the pre-training parameters and a unified fine-tuning paradigm, which narrow the gap between pre-training and fine-tuning and therefore embeds versatility capability on multiple benchmarks across different downstream tasks, including ISLR, CSLR, and SLT.

promising improvements in SLU tasks, they primarily focus on capturing visual cues from massive pre-training sign language data and lack joint modeling on the textual information, causing the gap with downstream task like SLT. To tackle this issue, some methods try to directly leverage video-gloss/video-text pairs to conduct pre-training, such as sign-to-gloss recognition (Chen et al., 2022b), video-text contrastive learning (Zhou et al., 2023), and pseudo-gloss prediction (Wong et al., 2024). Despite the incorporation of gloss/text data has been proven effective, they are generally limited by the scale of the video-gloss/video-text paired data or the transferring capability of downstream tasks.

To address these challenges, we introduce a unified pre-training framework that eliminates the gap between pre-training and downstream tasks, while operating at scale. As shown in Figure 1 (a,b), unlike previous pre-training methods, Uni-Sign utilizes generative pre-training on large-scale datasets, enforcing the model to capture the semantics embedded in sign language. Our approach consists of two key innovations. First, we introduce CSL-News, a large-scale Chinese Sign Language (CSL) dataset that spans 1,985 hours of videos with corresponding textual annotations, significantly surpassing existing CSL datasets in size and diversity. This dataset provides the foundation for large-scale pre-training. Second, we propose Uni-Sign, a pre-training model that unifies SLU tasks by treating downstream tasks as a single SLT task, ensuring seamless knowledge transfer between pre-training and fine-tuning. To further enhance performance, we integrate a prior-guided fusion (PGF) module and a score-aware sampling strategy, addressing keypoint inaccuracies and improving computational efficiency. In summary, our contributions are as follows

- We propose a unified pre-training framework, Uni-Sign, that achieves state-of-the-art performance across SLU tasks by eliminating the gap between pre-training and downstream tasks.
- We introduce CSL-News, a large-scale dataset with 1,985 hours of Chinese Sign Language videos and text annotations, enabling effective large-scale pre-training for SLU.
- We unify the pre-training and fine-tuning paradigm with shared objectives and incorporate a prior-guided fusion (PGF) module and a score-aware sampling strategy, which further improve performance by addressing keypoint inaccuracies and balancing speed with accuracy.

## 2 RELATED WORKS

### 2.1 SIGN LANGUAGE UNDERSTANDING

SLU encompasses various research fields, including ISLR, CSLR, and SLT.

**ISLR and CSLR.** ISLR focuses on classifying sign language movements. Previous works (Hu et al., 2021b; Li et al., 2020b; Zuo et al., 2023) have achieved superior performance by utilizing tailored models. CSLR aims to learn the sequence alignment between sign language and sign glosses, where each gloss is a manual transcription for a sign. Thanks to the capability of Connectionist Temporal Classification (CTC) loss (Graves et al., 2006) to effectively handle the alignment of two unsegmented sequences without precise alignment, it has become a mainstream approach in recent years (Pu et al., 2020; Min et al., 2021; Hu et al., 2023b; 2024; Jiao et al., 2023).

**SLT.** SLT requires the model to generate the corresponding text sequence by fully understanding sign language. It can be divided into two paradigms, gloss-based and gloss-free. By employing the gloss-based paradigm, the model acquires intermediate representation of glosses, leading to improved text generation capabilities. SLRT (Camgoz et al., 2020) pioneers the application of a transformer encoder-decoder framework in SLT and incorporates gloss-level supervision into the transformer encoder through CTC loss. STMC-T (Zhou et al., 2022) tackles the SLT through multi-cue modeling. SLTUNET (Zhang et al., 2023a) and MMTLB (Chen et al., 2022b) attempt to transfer knowledge from large-scale external text corpus and pre-trained language models into SLT to improve performance. However, the costly gloss labeling limits dataset and model scalability, prompting researchers to shift their attention toward the gloss-free paradigm. GFSLT-VLP (Zhou et al., 2023) novelly proposed text-video contrastive loss to pre-train translation models, which significantly boosted the performance of gloss-free methods. Sign2GPT (Wong et al., 2024) and Sign-LLM (Gong et al., 2024) aimed to take advantage of the linguistic knowledge inherent in large language models (LLMs) to enhance gloss-free SLT. In this paper, we also focus on gloss-free SLT, which is more challenging and easier to scale up in terms of both the model and the dataset.

Unlike the task-specific methods, we introduce a unified framework to handle these SLU tasks. Meanwhile, by eschewing any task-specific designs during the fine-tuning phase, our method maintains simplicity while consistently achieving remarkable performance across various SLU tasks.

## 2.2 SIGN LANGUAGE PRE-TRAINING

Sign language pre-training approaches leverage pretext tasks to capture semantic representations during the pre-training phase, resulting in notable performance improvements on diverse downstream tasks. Some researchers (Hu et al., 2021a; 2023a; Zhao et al., 2024b) attempt to leverage self-supervised learning to enhance representation capabilities from massive unlabeled data. Notably, the series of SignBERT (Hu et al., 2021a; 2023a) employs a masking-and-reconstructing strategy to mine contextual information of sign language, achieving promising performance improvements in SLU. However, these self-supervised sign language pre-training approaches primarily focus on learning low-level visual semantics while neglecting the acquisition of textual knowledge, resulting in a gap with downstream tasks such as SLT. Some studies have insightfully identified this issue and attempt to leverage video-gloss/video-text pair data to inject linguistic knowledge into the pre-trained model. MMTLB (Chen et al., 2022b) achieves precise alignment between sign language and text by employing three sub-tasks (sign-to-gloss, gloss-to-text, and sign-to-text), thereby unlocking the potential of pre-trained language models. GFSLT-VLP (Zhou et al., 2023) proposes a contrastive learning pretext task that effectively aligns sign language and text in a joint space, significantly advancing the development of gloss-free SLT. Inspired by GFSLT-VLP, MSLU (Zhou et al., 2024) and $C^2RL$ (Chen et al., 2024a) further introduce the pretext tasks of keypoint reconstruction and language modeling, respectively. Despite the effectiveness of incorporating gloss/text data, they are generally limited by the scale of the video-gloss/video-text paired data or the transferring capability of downstream tasks. YouTube-ASL (Uthus et al., 2024) directly employs language modeling task for large-scale pre-training, demonstrating the potential of generative pre-training and emphasizing the importance of scaling datasets. In contrast to prior pre-training methods (Hu et al., 2021a; 2023a; Zhao et al., 2023; Zhou et al., 2023), we propose a framework that benefits from large-scale pre-training and a unified pre-training and fine-tuning paradigm, thereby fully unlocking the SLU potential of the pre-trained model and transferring it to downstream tasks.

## 2.3 UNIFYING VIA LANGUAGE MODELING

Inspired by the success of sequence-to-sequence (seq2seq) modeling in natural language processing, previous studies (Chen et al., 2022a; Wang et al., 2022) employ seq2seq approaches to unify a vari-

| Name | Language | Vocab. | Hours | Source |
|------|----------|--------|-------|--------|
| KETI (Ko et al., 2019) | KVK | 419 | 28 | Lab |
| SWISSTXT (Camgöz et al., 2021) | DSGS | - | 88 | TV |
| VRT-RAW (Camgöz et al., 2021) | VGT | - | 100 | TV |
| PHOENIX-2014T (Camgoz et al., 2018) | DGS | 3K | 11 | TV |
| DGS Corpus (Hanke et al., 2020) | DGS | - | 50 | Lab |
| BOBSL (Albanie et al., 2021) | BSL | 77K | 1,447 | TV |
| How2Sign (Duarte et al., 2021) | ASL | 16K | 79 | Lab |
| OpenASL (Shi et al., 2022) | ASL | 33K | 288 | Web |
| YouTube-ASL (Uthus et al., 2024) | ASL | 60K | 984 | Web |
| SP-10 (Yin et al., 2022) | various | 17K | 14 | Web |
| AfriSign (Gueuwou et al., 2023b) | various | 20K | 152 | Web |
| CSL-Daily (Zhou et al., 2021) | CSL | 2K | 23 | Lab |
| CSL-News (Ours) | CSL | 5K | 1,985 | TV |

Table 1: Summary statistics for different SLT datasets.

Figure 2: Distribution of video durations and text lengths.

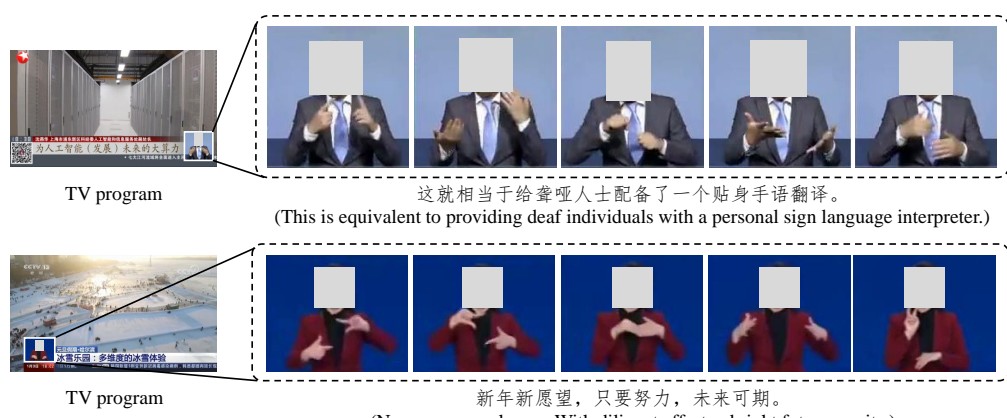

这就相当于给聋哑人士配备了一个贴身手语翻译。
(This is equivalent to providing deaf individuals with a personal sign language interpreter.)

新年新愿望，只要努力，未来可期。
(New year, new hopes. With diligent effort, a bright future awaits.)

Figure 3: Samples of videos and text annotations in the CSL-news dataset. The signer's face is masked in here to protect their privacy.

ety of tasks. Built on these advancements, vision LLMs (Li et al., 2023; Zhu et al., 2024) extend the capabilities of LLMs to vision-language understanding by leveraging language modeling objective. LLaVA (Liu et al., 2023) demonstrates remarkable multimodal instruction-following capabilities by utilizing the extensive world knowledge embedded in LLMs. VisionLLM-v2 (Wu et al., 2024) utilizes language modeling to tackle hundreds of vision-language tasks, further highlighting the effectiveness of the unified paradigm. Motivated by these advancements, we propose Uni-Sign, which aims to address various SLU tasks via language modeling, while also achieving both simplicity and scalability.

## 2.4 SIGN LANGUAGE UNDERSTANDING DATASETS

Collecting large-scale and high-quality datasets is crucial for improving neural network performance and has been widely explored. For ISLR, various benchmarks (Joze & Koller, 2019; Li et al., 2020a; Hu et al., 2021c) have been proposed to comprehensively evaluate model performance. Phoenix2014-T (Camgoz et al., 2018) CSL-Daily (Zhou et al., 2021) are introduced to tackle CSLR and SLT tasks. Although numerous efforts (Shi et al., 2022; Duarte et al., 2021; Yin et al., 2022; Hanke et al., 2020) have been made to develop high-quality datasets to boost SLU, the field is still constrained by the size of the available datasets. BoBSL (Albanie et al., 2021) and YouTube-ASL (Uthus et al., 2024) insightfully identified this issue and proposed a 1,447 hours British Sign Language (BSL) dataset and a 984 hours American Sign Language (ASL) dataset, respectively. Additionally, YouTube-SL-25 (Tanzer & Zhang, 2024) and JWSign (Gueuwou et al., 2023a) have also collected large-scale multilingual sign language datasets, which are crucial for training unified multilingual sign language models. Previous works primarily focused on collecting BSL (Albanie et al., 2021) and ASL (Shi et al., 2022; Uthus et al., 2024; Tanzer & Zhang, 2024) datasets, leaving

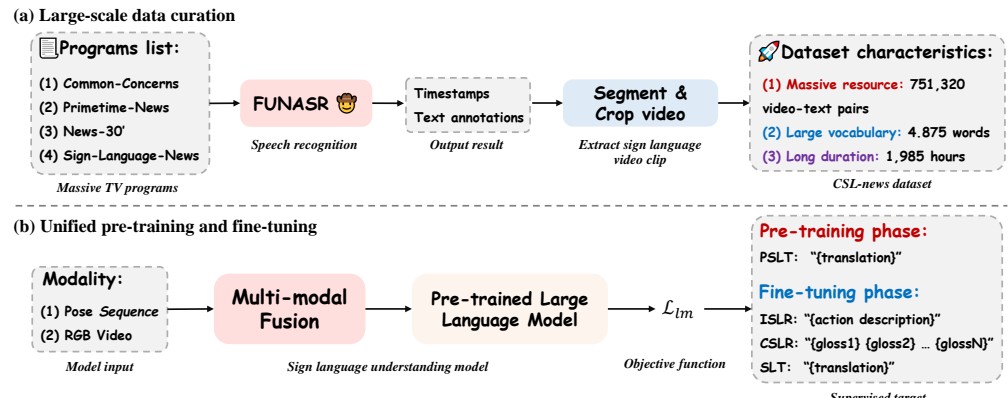

Figure 4: The overview of our two key innovations: (a) Pipeline for large-scale data curation. (b) Unified pre-training and fine-tuning, utilizing pre-training parameters and a single language modeling loss to address diverse SLU tasks.

CSL datasets relatively underexplored. To fill this gap, we propose CSL-News, a 1,985 hours CSL translation dataset.

## 3 METHOD

### 3.1 LARGE-SCALE DATA CURATION: CSL-NEWS

Currently, the larger publicly available SLT datasets are mainly sourced from ASL (Shi et al., 2022; Uthus et al., 2024; Tanzer & Zhang, 2024) and BSL (Albanie et al., 2021), there still exists an urgent need to collect a large-scale CSL dataset. As illustrated in Table 1, CSL-Daily (Zhou et al., 2021) is currently the largest existing CSL dataset which only contains a total duration of 23 hours and is insufficient to train a robust CSL model. We therefore gather the CSL-News dataset, a large-scale SLT dataset with 1,985 hours of videos, approximately 86 times larger than the CSL-Daily dataset.

To construct this dataset, we primarily collect four TV programs[1] from three different TV station to construct our dataset. The duration statistics for each TV station are as follows: CCTV-13, 1,342 hours; Dragon TV, 623 hours; and Hebei Radio and TV Station, 20 hours. After downloading the massive TV programs and considering the strong temporal alignment between sign language and news broadcasts, we employ the FunASR (Gao et al., 2023) toolkit to extract textual annotations from the audio. Subsequently, the news videos are segmented based on the timestamps of punctuation marks ( 。, ？, ！ ) to generate video-text pairs. Finally, we crop the sign language videos using predefined relative coordinates to eliminate background interference. Through these processes, we curate a large-scale CSL translation dataset that plays a crucial role in the pre-training of a large-scale CSL model. As shown in Figure 2, the dataset comprises video clips with an average duration of 9.5 seconds and an average text length of 40 words (Chinese characters) in a total of 751,320 video clips. However, in this paper, we utilize only 722,711 video clips shorter than 512 frames for training. More discussion could be found in Appendix A.5. Figure 3 illustrates the videos and text annotations within the CSL-News dataset, while Figure 4 (a) presents the complete pipeline for large-scale data curation.

### 3.2 UNIFIED PRE-TRAINING AND FINE-TUNING

For training efficiency, the process is divided into three stages, Stage 1: pose-only pre-training, Stage 2: RGB-pose interaction continue pre-training, and Stage 3: downstream task fine-tuning. The framework of Uni-Sign is illustrated in Figure 5 (a).

**Preliminaries.** Given 133 keypoints, we selectively utilize 69 keypoints, categorizing them into three sub-pose groups: 21 for each hand, 9 for the body, and 18 for the face. The keypoints sequence

---

[1]Common-Concerns, Primetime-News, News-30', Sign-Language-News

of group $i$ is then processed by its corresponding pose encoder, producing gesture features $\mathcal{F}_i^p \in \mathbb{R}^{T \times N_i \times C}$. Here, $T$ represents the length of the pose sequence, $N_i$ denotes the number of keypoints in group $i$, $C$ is the dimension of the features, and $i \in \{lh, rh, b, f\}$. In this paper, pose encoder of group $i$ is composed of a three-layer spatial GCN.

Following the idea of decoupling visual cues, the vision encoder focuses on learning representations from both hands rather than the entire image. To achieve this, videos cropped using keypoint coordinates and resized to $112 \times 112$ pixels are processed by the vision encoder, producing vision features denoted as $\mathcal{F}_{lh}^r \in \mathbb{R}^{T \times h \times w \times C}$ and $\mathcal{F}_{rh}^r \in \mathbb{R}^{T \times h \times w \times C}$. Notably, in Stage 2, we fuse $\mathcal{F}_i^p$ and $\mathcal{F}_i^r$ to compensate for the visual information lost due to inaccurate keypoints, obtaining the fused features $\tilde{\mathcal{F}}_i^p$. Specific details of the fusion module will be provided in Section 3.3. After those processes, we employ a three-layer ST-GCN (Yan et al., 2018) to construct the short-term temporal encoder. The features from each group ($\mathcal{F}_i^p$ or $\tilde{\mathcal{F}}_i^p$) are then fed into the temporal encoders, aggregated intra-group via a mean pooling layer, and concatenated across all groups to produce the final feature $\mathcal{F}_{sign} \in \mathbb{R}^{T \times 4C}$, which is subsequently input to the language model.

**Pre-training Uni-Sign.** Previous works (Chen et al., 2022b;c; Zhao et al., 2024a; Wong et al., 2024; Chen et al., 2024b; Zhou et al., 2023) have designed indirect pretext tasks (e.g., gloss-to-text translation, pseudo-gloss prediction) to unlock the pre-trained language model's potential. Different from them, we directly employ the generative pre-training paradigm to utilize the knowledge embedded within the pre-trained large language model. Specifically, we project the feature $\mathcal{F}_{sign}$ to match the dimension of the language model and then feed it into the language model. The loss function is as follows:

$$\mathcal{L}_{lm} = -\sum_{u=1}^{U} \log p(s_u | s_{<u}, \mathcal{F}_{sign}), \tag{1}$$

where $s_u$ represents the $u$-th token, and $s_{<u}$ denotes all preceding tokens in the sentence $s$. During the pre-training phase, we leverage $\mathcal{L}_{lm}$ as the objective function, denoted as $\mathcal{L}_{\text{PSLT}}$, as depicted in Figure 4 (a).

**Fine-tuning Uni-Sign.** Although there are specific fine-tuning methods tailored to each task (e.g., ISLR employs an MLP head for classification, while CSLR commonly utilizes CTC loss to enforce temporal constraints), we innovatively treat ISLR, CSLR, and SLT as a single SLT task, allowing us to employ a unified fine-tuning paradigm to fine-tune all SLU tasks without the bells-and-whistles, as shown in Figure 4 (b). To construct supervision targets, ISLR uses action description, CSLR employs sequences of glosses separated by spaces, and SLT utilizes the translation text, denoted as $y_{\text{word}}$, $y_{\text{gloss}}$, $y_{\text{sentence}}$, respectively. Through this setting, the robust SLU capabilities integrated into the model during the large-scale pre-training phase will be seamlessly transferred to downstream tasks, thereby unlocking the full potential of the pre-trained model.

In summary, the objective function of each phase are as follows:

$$\text{Pre-training} \left\{ \mathcal{L}_{\text{PSLT}} = \mathcal{L}_{lm}(\mathcal{F}_{sign}, y_{\text{sentence}}) \right.$$

$$\text{Fine-tuning} \left\{ \begin{array}{l} \mathcal{L}_{\text{ISLR}} = \mathcal{L}_{lm}(\mathcal{F}_{sign}, y_{\text{word}}) \\ \mathcal{L}_{\text{CSLR}} = \mathcal{L}_{lm}(\mathcal{F}_{sign}, y_{\text{gloss}}) \\ \mathcal{L}_{\text{SLT}} = \mathcal{L}_{lm}(\mathcal{F}_{sign}, y_{\text{sentence}}) \end{array} \right. \tag{2}$$

### 3.3 MULTI-MODAL FUSION

**Prior-guided fusion.** Multimodal networks (Jiang et al., 2021; Chen et al., 2022c; Zuo et al., 2023; Jiang et al., 2024) have been widely explored in SLU. However, most existing methods simply perform spatial-temporal fusion (e.g., concatenation, cross-attention) without considering the fine-grained spatial relationships, which are crucial for narrowing the representational gap between modalities. Hence, we propose a prior-guided fusion (PGF) module that leverages keypoint coordinates as priors to model fine-grained spatial consistency between modalities, as illustrated in Figure 5 (b). Given $\mathcal{F}_{i,t}^p$ and $\mathcal{F}_{i,t}^r$, where $i = \{lh, rh\}$, we first employ a multi-head attention module to incorporate the global RGB information into $\mathcal{F}_{i,t}^p$. Then, by utilizing the keypoint coordinates $J_{i,t}^s$ as priors to initialize the reference points in deformable attention (Xia et al., 2022), fine-grained spatial

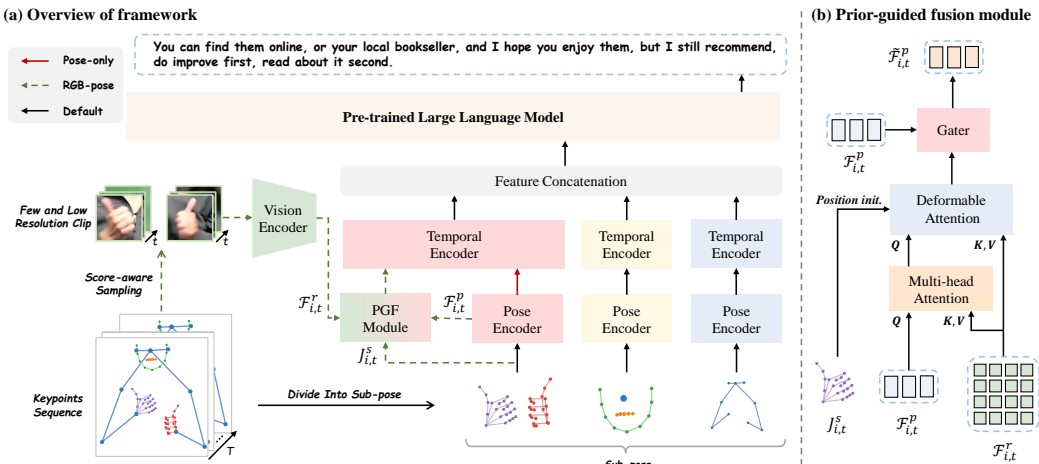

Figure 5: (a) The framework of Uni-Sign. In the pose-only setting, the keypoints are divided into sub-pose (hands, face, and body) and fed into pose encoders and temporal encoders to capture the fine-grained visual cue. Subsequently, features from each part at the same time step are concatenated along the feature dimension and processed by a pre-trained large language model to generate text. In the RGB-pose setting, a score-aware sampling strategy is introduced to sample low-confidence frames and crop the corresponding hand regions. The hands are further encoded by a vision encoder, interacting with hand pose features through a PGF module to mitigate the impact of inaccurate keypoints. (b) The overview of PGF module, which fuses RGB and pose features frame by frame.

modeling across modalities is achieved. The fused features are denoted as $\hat{\mathcal{F}}^p_{i,t}$. Finally, $\mathcal{F}^p_{i,t}$ and $\hat{\mathcal{F}}^p_{i,t}$ are fed into a gater to expedite convergence during Stage 2 training. The implementation details are as follows:

$$g = \text{Gate}([\mathcal{F}^p_{i,t}, \hat{\mathcal{F}}^p_{i,t}]), \tag{3}$$

$$\tilde{\mathcal{F}}^p_{i,t} = (1 - g) * \mathcal{F}^p_{i,t} + g * \hat{\mathcal{F}}^p_{i,t}, \tag{4}$$

where Gate is a gate module initialized to zero, aimed to preserve the knowledge learned in Stage 1 at the beginning of Stage 2. The notation $[\cdot, \cdot]$ indicates the concatenation operation. Although $\{\mathcal{F}^p_b, \mathcal{F}^p_f\}$ are not fused with RGB, we also convert their notation to $\{\tilde{\mathcal{F}}^p_b, \tilde{\mathcal{F}}^p_f\}$ for ease of expression.

**Score-aware sampling strategy.** Although RGB-pose fusion compensates for the visual cues lost due to inaccurate keypoints, the inclusion of RGB poses a significant challenge to computational resources. Considering the information redundancy between RGB and high-confidence keypoints, we propose a score-aware sampling strategy that selectively chooses RGB frames corresponding to low-confidence keypoints, thus balancing performance and speed. To this end, we use the average confidence of hand keypoints as the reliability score $rs$, subsequently calculating the sampling score as $1 - rs$. Next, we randomly sample $P_{samp}\%$ of RGB frames based on these sampling scores. Finally, by employing indexing, the sampled RGB frames are efficiently interacted with their corresponding pose features. The relevant pseudocode is presented in Appendix A.3.

## 4 EXPERIMENTS

### 4.1 IMPLEMENTATION DETAILS

For Stage 1 and Stage 2, we utilize CSL-News and YouTube-ASL (Uthus et al., 2024) as pre-training datasets for CSL and ASL, respectively. In Stage 3, fine-tuning is conducted separately for each downstream dataset. We implement Uni-Sign using PyTorch (Paszke et al., 2019), employing mT5-Base (Xue et al., 2021) as our pre-trained language

| Config | Stage 1 | Stage 2 | Stage 3 |
|---|---|---|---|
| optimizer | | AdamW | |
| base learning rate | | 3e-4 | |
| weight decay | | 1e-4 | |
| optimizer momentum | | $\beta_1, \beta_2 = 0.9, 0.999$ | |
| learning rate schedule | | cosine decay | |
| training epochs | 20 | 5 | 20 |
| batch size | 16 | 4 | 8 |
| gradient accumulation | 8 | 8 | 1 |

Table 2: Training recipe of each stage.

| Method | Modality | | MSASL100 | | MSASL1000 | | WLASL100 | | WLASL2000 | |
|---|---|---|---|---|---|---|---|---|---|---|
| | Pose | RGB | P-I | P-C | P-I | P-C | P-I | P-C | P-I | P-C |
| ST-GCN† (Yan et al., 2018) | ✔ | | 50.78 | 51.62 | 34.40 | 32.53 | 50.78 | 51.62 | 34.40 | 32.53 |
| SignBERT (Hu et al., 2021a) | ✔ | | 76.09 | 76.65 | 49.54 | 46.39 | 76.36 | 77.68 | 39.40 | 36.74 |
| BEST (Zhao et al., 2023) | ✔ | | 80.98 | 81.24 | 58.82 | 54.87 | 77.91 | 77.83 | 46.25 | 43.52 |
| SignBERT+ (Hu et al., 2023a) | ✔ | | 84.94 | 85.23 | 62.42 | 60.15 | 79.84 | 80.72 | 48.85 | 46.37 |
| MSLU (Zhou et al., 2024) | ✔ | | 91.54 | 91.75 | 74.07 | 71.81 | 88.76 | 89.25 | 56.29 | 53.29 |
| HMA (Hu et al., 2021b) | | ✔ | 73.45 | 74.59 | 49.16 | 46.27 | - | - | 37.91 | 35.90 |
| TCK (Li et al., 2020b) | | ✔ | 83.04 | 83.91 | - | - | 77.52 | 77.55 | - | - |
| NLA-SLR (Zuo et al., 2023) | ✔ | ✔ | 90.49 | 91.04 | 72.56 | 69.86 | 91.47 | 92.17 | 61.05 | 58.05 |
| Uni-Sign (Ours) | ✔ | | 93.26 | 93.16 | 77.88 | 76.55 | 92.24 | 92.75 | 63.13 | 60.90 |
| Uni-Sign (Ours) | ✔ | ✔ | 93.79 | 94.02 | 78.16 | 76.97 | 92.25 | 92.67 | 63.52 | 61.32 |

Table 3: ISLR results on various benchmarks. † denotes methods reproduced by (Hu et al., 2021a). Blue and Green denote the best results of previous methods and ours, respectively.

| Method | Modality | | CSL-Daily | |
|---|---|---|---|---|
| | Pose | RGB | Dev | Test |
| MSLU (Zhou et al., 2024) | ✔ | | 28.6 | 27.9 |
| CoSign (Jiao et al., 2023) | ✔ | | 28.1 | 27.2 |
| SignBT (Zhou et al., 2021) | | ✔ | 33.2 | 33.2 |
| AdaBrowse (Hu et al., 2023d) | | ✔ | 31.2 | 30.7 |
| SEN (Hu et al., 2023c) | | ✔ | 31.1 | 30.7 |
| CorrNet (Hu et al., 2023b) | | ✔ | 30.6 | 30.1 |
| C2ST (Zhang et al., 2023b) | | ✔ | 25.9 | 25.8 |
| TS-SLR (Chen et al., 2022c) | ✔ | ✔ | 25.4 | 25.3 |
| Uni-Sign (Ours) | ✔ | | 28.2 | 27.4 |
| Uni-Sign (Ours) | ✔ | ✔ | 26.7 | 26.0 |

Table 4: CSLR results on CSL-Daily dataset with WER scores.

| Method | Modality | | Dev | | | Test | | |
|---|---|---|---|---|---|---|---|---|
| | Pose | RGB | BLEU1 | BLEU4 | ROUGE | BLEU1 | BLEU4 | ROUGE |
| *Gloss-based* | | | | | | | | |
| SLRT† (Camgoz et al., 2020) | | ✔ | 37.47 | 11.88 | 37.96 | 37.38 | 11.79 | 36.74 |
| ConSLT (Fu et al., 2023) | | ✔ | - | 14.80 | 41.46 | - | 14.53 | 40.98 |
| SignBT (Zhou et al., 2021) | | ✔ | 51.46 | 20.80 | 49.49 | 51.42 | 21.34 | 49.31 |
| SLTUNET (Zhang et al., 2023a) | | ✔ | - | 23.99 | 53.58 | 54.98 | 25.01 | 54.08 |
| MMTLB (Chen et al., 2022b) | | ✔ | 53.81 | 24.42 | 53.38 | 53.31 | 23.92 | 53.25 |
| CV-SLT (Zhao et al., 2024a) | | ✔ | - | 28.24 | 56.36 | 58.29 | 28.94 | 57.06 |
| TS-SLT (Chen et al., 2022c) | ✔ | ✔ | 55.21 | 25.76 | 55.10 | 55.44 | 25.79 | 55.72 |
| *Gloss-free* | | | | | | | | |
| MSLU (Zhou et al., 2024) | ✔ | | 33.28 | 10.27 | 33.13 | 33.97 | 11.42 | 33.80 |
| SLRT‡ (Camgoz et al., 2020) | | ✔ | 21.03 | 4.04 | 20.51 | 20.00 | 3.03 | 19.67 |
| GASLT (Yin et al., 2023) | | ✔ | - | - | - | 19.90 | 4.07 | 20.35 |
| NSLT† (Camgoz et al., 2018) | | ✔ | 34.22 | 7.96 | 34.28 | 34.16 | 7.56 | 34.54 |
| GFSLT-VLP (Zhou et al., 2023) | | ✔ | 39.20 | 11.07 | 36.70 | 39.37 | 11.00 | 36.44 |
| FLa-LLM (Chen et al., 2024b) | | ✔ | - | - | - | 37.13 | 14.20 | 37.25 |
| Sign2GPT (Wong et al., 2024) | | ✔ | - | - | - | 41.75 | 15.40 | 42.36 |
| SignLLM (Gong et al., 2024) | | ✔ | 42.45 | 12.23 | 39.18 | 39.55 | 15.75 | 39.91 |
| C2RL (Chen et al., 2024a) | | ✔ | - | - | - | 49.32 | 21.61 | 48.21 |
| Uni-Sign (Ours) | ✔ | | 53.24 | 25.27 | 54.34 | 53.86 | 25.61 | 54.92 |
| Uni-Sign (Ours) | ✔ | ✔ | 55.30 | 26.25 | 56.03 | 55.08 | 26.36 | 56.51 |

Table 5: SLT results on CSL-Daily dataset. † and ‡ denote methods reproduced by (Zhou et al., 2021) and (Zhou et al., 2023), respectively. Underline indicates the best gloss-based SLT result.

| Method | Modality | | Test | | | |
|---|---|---|---|---|---|---|
| | Pose | RGB | BLEU1 | BLEU4 | ROUGE | BLEURT |
| *How2Sign* | | | | | | |
| GloFE-VN (Lin et al., 2023) | ✔ | | 14.9 | 2.2 | 12.6 | 31.7 |
| YouTube-ASL (Uthus et al., 2024) | ✔ | | 37.8 | 12.4 | - | 46.6 |
| MSLU (Zhou et al., 2024) | ✔ | | 20.1 | 2.4 | 17.2 | - |
| SLT-IV (Tarrés et al., 2023) | | ✔ | 34.0 | 8.0 | - | - |
| C2RL (Chen et al., 2024a) | | ✔ | 29.1 | 9.4 | 27.0 | - |
| FLa-LLM (Chen et al., 2024b) | | ✔ | 29.8 | 9.7 | 27.8 | - |
| SignMusketeers (Gueuwou et al., 2024) | | ✔ | 41.5 | 14.3 | - | - |
| SSVP-SLT (Rust et al., 2024) | | ✔ | 43.2 | 15.5 | 38.4 | 49.6 |
| Uni-Sign (Ours) | ✔ | | 40.4 | 14.5 | 34.3 | 48.6 |
| Uni-Sign (Ours) | ✔ | ✔ | 40.2 | 14.9 | 36.0 | 49.4 |
| *OpenASL* | | | | | | |
| GloFE-VN (Lin et al., 2023) | ✔ | | 21.56 | 7.06 | 21.75 | 36.35 |
| Conv-GRU† (Camgoz et al., 2018) | | ✔ | 16.11 | 4.58 | 16.10 | 25.65 |
| I3D-transformer (Shi et al., 2022) | | ✔ | 18.31 | 5.66 | 18.64 | 28.82 |
| OpenASL (Shi et al., 2022) | | ✔ | 20.92 | 8.59 | 21.02 | 31.09 |
| C2RL (Chen et al., 2024a) | | ✔ | 31.46 | 13.21 | 31.36 | - |
| Uni-Sign (Ours) | ✔ | | 49.10 | 22.67 | 42.77 | 60.08 |
| Uni-Sign (Ours) | ✔ | ✔ | 49.35 | 23.14 | 43.22 | 60.40 |

Table 6: Gloss-free SLT results on How2Sign and OpenASL. † indicates methods reproduced by (Shi et al., 2022).

model. The mT5-Base model benefits from pre-training on the mC4 (Xue et al., 2021) corpus, which enhances its multilingual understanding capabilities. Additionally, the vision encoder is an EfficientNet-B0 (Tan & Le, 2019) pre-trained on ImageNet (Deng et al., 2009). We did not use any data augmentation during training. The detailed training recipe is presented in Table 2.

## 4.2 DATASETS AND EVALUATION METRICS

**Datasets.** We evaluate our model on various benchmarks to demonstrate its effectiveness. For ISLR, we adopt WLASL (Li et al., 2020a) and MSASL (Joze & Koller, 2019) datasets for evaluation. For CSLR, we utilize CSL-Daily (Zhou et al., 2021). SLT task is conducted on the CSL-Daily, How2Sign (Duarte et al., 2021), and OpenASL (Shi et al., 2022) datasets.

**Evaluation metrics.** Following previous works (Hu et al., 2021a; Zhou et al., 2024), we report per-instance (P-I) and per-class (P-C) Top-1 accuracy, as well as word error rate (WER), to evaluate ISLR and CSLR, respectively. For SLT, we adopt BLEU (Papineni et al., 2002) from the SacreBLEU (Post, 2018) library and ROUGE-L (Lin, 2004) as evaluation metrics. For English SLT datasets, we also report BLEURT (Sellam et al., 2020) scores using the BLEURT-20 checkpoint, as it has been shown to correlate strongly with human judgments.

## 4.3 COMPARISON WITH STATE-OF-THE-ART METHODS

To validate the effectiveness of our framework, we conduct a series of experiments across a diverse range of SLU tasks. To provide additional references for future research, we present both the performance of the RGB-pose setting and the pose-only setting, where the pose-only experiments bypass the training in Stage 2 entirely. Due to page limitations, the qualitative visualization is presented in Appendix A.4.

**Results on ISLR and CSLR.** We compare the results of Uni-Sign with previous studies on ISLR benchmarks in Table 3. Our model surpasses previous SOTA on these benchmarks without any task-specific designs. Compared to the previous state-of-the-art methods (MSLU and NLA-SLR), our approach achieves improvements of 4.09% and 2.47% in per-instance Top-1 accuracy on the

| Setting | MSASL1000 (ISLR) | | CSL-Daily (CSLR) | |
|---|---|---|---|---|
| | P-I Top-1↑ | P-C Top-1↑ | Dev WER↓ | Test WER↓ |
| Task-specific fine-tuning paradigm | | | | |
| $\mathcal{F}_{sign}$ | 56.92 | 53.67 | 37.4 | 36.4 |
| $\mathcal{F}_{lm\_enc}$ | 70.97 | 68.54 | 39.2 | 38.3 |
| Unified fine-tuning paradigm | | | | |
| Ours | 77.88 | 76.55 | 28.2 | 27.4 |

Table 7: Impact of fine-tuning paradigm in pose-only setting.

| Setting | CSL-Daily (CSLR) | | CSL-Daily (SLT) | | | |
|---|---|---|---|---|---|---|
| | Dev WER↓ | Test WER↓ | Dev BLEU4↑ | Dev ROUGE↑ | Test BLEU4↑ | Test ROUGE↑ |
| 0% | 74.7 | 73.6 | 3.75 | 20.46 | 3.51 | 20.56 |
| 25% | 31.5 | 31.0 | 20.68 | 49.68 | 21.13 | 49.9 |
| 50% | 31.5 | 30.1 | 21.85 | 50.98 | 22.58 | 51.62 |
| 75% | 28.8 | 28.5 | 24.74 | 54.28 | 24.95 | 54.87 |
| 100% | **28.2** | **27.4** | **25.27** | **54.34** | **25.61** | **54.92** |

Table 8: Impact of pre-training data scale in pose-only setting.

| $P_{samp}$ | Time | CSL-Daily (CSLR) | | CSL-Daily (SLT) | | | |
|---|---|---|---|---|---|---|---|
| | | Dev WER↓ | Test WER↓ | Dev BLEU4↑ | Dev ROUGE↑ | Test BLEU4↑ | Test ROUGE↑ |
| 0 % | 1.0× | 28.2 | 27.4 | 25.27 | 54.34 | 25.61 | 54.92 |
| 10 % | 1.3× | **26.7** | **26.0** | 26.25 | 56.03 | 26.36 | 56.51 |
| 25 % | 1.7× | 27.0 | 26.2 | **26.37** | 56.11 | 26.55 | 56.48 |
| 50 % | 2.2× | 26.8 | 26.4 | 26.30 | **56.39** | **26.86** | **57.43** |

Table 9: Impact of score-aware sampling strategy in RGB-pose setting.

| Setting | CSL-Daily (CSLR) | | CSL-Daily (SLT) | | | |
|---|---|---|---|---|---|---|
| | Dev WER↓ | Test WER↓ | Dev BLEU4↑ | Dev ROUGE↑ | Test BLEU4↑ | Test ROUGE↑ |
| CA | 27.3 | 27.0 | 25.74 | 55.63 | 26.11 | 56.22 |
| DA | **26.7** | **26.0** | **26.30** | **56.03** | **26.36** | **56.51** |

Table 10: Impact of fusion module in RGB-pose setting.

challenging MSASL1000 and WLASL2000 datasets, respectively. Moreover, we evaluate the performance on CSLR, as shown in Table 4. Despite not employing CTC loss to impose temporal constraints on sign language, our model still shows competitive performance, with only a 1.3% and 0.7% performance drop compared to TS-SLR. We argue that the performance gap between Uni-Sign and TS-SLR may be attributed to the more complex model architecture and the dense intermediate state constraints incorporated in TS-SLR. The experiments demonstrate that our approach learns robust SLU capabilities via pre-training and successfully transfers the knowledge through generative fine-tuning to downstream tasks.

**Results on SLT.** Table 5 and 6 present comparisons of SLT performance between our method and prior approaches on the CSL-Daily, How2Sign, and OpenASL datasets. We observe that Uni-Sign beats previous gloss-free SOTA on CSL-Daily and OpenASL, with a substantial performance increase in BLEU4. On the CSL-Daily dataset, under the same gloss-free paradigm, Uni-Sign surpasses previous SOTA, achieving improvements of 14.02 and 4.75 in the BLEU4 scores on the dev and test sets, respectively. Moreover, we surprisingly found that Uni-Sign also outperforms certain gloss-based SLT models, such as SLTUNET and TS-SLT, which integrated gloss information into their frameworks through CSLR training. The series of results above emphasizes the importance of large-scale generative pre-training, which endows the model with robust SLU capabilities. On the OpenASL dataset, our method outperforms C[2]RL by 9.93 in BLEU4 and achieves a remarkable BLEURT score (60.40 vs. 36.35). Meanwhile, our performance on How2Sign is also notable, achieving comparable results with RGB-based models (SignMusketeers, SSVP-SLT) under the same pre-training dataset. Although SSVP-SLT employs a larger-scale vision encoder, a more complex pre-training strategy, and a longer pre-training duration, Uni-Sign demonstrates only a slight performance difference in terms of BLEU4 (14.9 vs. 15.5), highlighting its competitive potential.

## 4.4 ABLATION STUDY

We conduct various ablation studies to investigate the contribution of each key component in Uni-Sign. Specifically, the MSASL1000 dataset is selected for ISLR, while the CSL-Daily dataset is used for the other tasks.

**Impact of fine-tuning paradigms.** We separately utilize features from the temporal encoders and the language model encoder as the representations of sign language, denoted as $\mathcal{F}_{sign}$ and $\mathcal{F}_{lm\_enc}$, respectively. These features are then performed task-specific fine-tuning settings to investigate the impact of different fine-tuning paradigms. For ISLR, the selected features undergo mean pooling followed by a classification head, supervised by cross-entropy loss. For CSLR, the features are passed through an LSTM layer and optimized by CTC loss. As depicted in Table 7, our proposed fine-tuning paradigm achieves the best performance, demonstrating a notable margin over task-specific fine-tuning paradigms. We also observe that the features $\mathcal{F}_{lm\_enc}$ yield better results in ISLR than $\mathcal{F}_{sign}$, while performing worse in CSLR. This suggests that while $\mathcal{F}_{lm\_enc}$ captures high-level semantics of sign language, it compromises short-term temporal understanding. Despite

the rich semantic information encoded in $\mathcal{F}_{lm\_enc}$, an improper fine-tuning method resulted in a performance drop of 6.91% for P-I and 8.01% for P-C in ISLR. Furthermore, the unified fine-tuning paradigm leverages its robust SLU understanding and linguistic restructuring capabilities, significantly outperforming the mainstream CSLR fine-tuning paradigm that uses CTC loss for temporal constraints (28.2 vs. 37.4 and 27.4 vs. 36.4). The above results highlight that our proposed unified fine-tuning paradigm can effectively transfer the SLU capabilities within the pre-trained model.

**Impact of pre-training data scale.** We randomly sample a portion of data from the CSL-News dataset for pre-training to explore the impact of pre-training data scale on model performance. In Table 8, we observe that as the quantity of pre-training data increases, the performance of various tasks progressively improves, indicating that our model can benefit from larger datasets and highlighting the critical role of large-scale pre-training.

**Impact of score-aware sampling strategy.** To evaluate the effectiveness of the score-aware sampling strategy, we perform hyperparameter selection on the sampling probability $P_{samp}$. As illustrated in Table 9, increasing $P_{samp}$ results in a gradual improvement in SLT, achieving a maximum gain of 1.25 BLEU4 on the CSL-Daily test set when $P_{samp}$ reaches 50%. However, time consumption also increases significantly. To balance performance and time consumption, we select 10% as the default value, which still yields promising results.

**Impact of fusion module.** To evaluate the impact of the fusion module, we replace the deformable attention (DA) in the PGF module with cross-attention (CA). The results presented in Table 10 demonstrate that DA outperforms CA in the CSLR, and SLT tasks, highlighting the effectiveness of using keypoint coordinates as priors.

## 5 CONCLUSION AND FUTURE WORK

In this paper, we introduce Uni-Sign, a unified pre-training framework that leverages a large-scale generative pre-training strategy and a novel fine-tuning paradigm, narrowing the gap between pre-training and downstream SLU tasks. Specifically, we first propose CSL-News, a large-scale CSL translation dataset containing 1,985 hours of video-text pairs, which enables effective large-scale pre-training. Next, we unify the fine-tuning paradigm by treating downstream SLU tasks as a single SLT task, which significantly narrows the gap between pre-training and fine-tuning while facilitating the transfer of SLU capabilities to these tasks. Moreover, we introduce the PGF module and a score-aware sampling strategy to efficiently capture visual cues from both RGB and pose modalities while achieving a trade-off between performance and speed. Despite the simplicity of Uni-Sign, we achieve remarkable results across multiple SLU tasks, demonstrating a notable improvement over previous state-of-the-art methods.

In the future, we are interested in exploring large-scale pre-trained multilingual SLU models and SLU tasks in complex scenarios (such as complex backgrounds, multi-signer situations, long-duration sign language understanding). We are also keen to investigate sign language production, a research field as important as SLU, to ensure that the Deaf/Hard of Hearing communities can equally benefit from technological advancements.

## REPRODUCIBILITY STATEMENT

To facilitate reproducibility, we have provided details of the training settings in Section 4.1, with further details of our framework to be presented in Appendix A.1. The CLS-News dataset and the code of Uni-Sign have been open-sourced and made available, aiming to promote further research on SLU.

## ACKNOWLEDGMENTS

This work was supported by National Natural Science Foundation of China under Contract U20A20183 & 62021001, and the Youth Innovation Promotion Association CAS. It was also supported by the GPU cluster built by MCC Lab of Information Science and Technology Institution, USTC, and the Supercomputing Center of the USTC.

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

# A APPENDIX

## A.1 FRAMEWORK IMPLEMENTATION

**Keypoints extraction.** We employ the RTMPose-x (Jiang et al., 2023) from MMPose to extract whole-body keypoints. The visualization of whole-body keypoints are shown in Figure 6. As mentioned in Section 3.2, we divide the pose into sub-pose (left hand, right hand, face, and body). We select the indices for the left hand ({92-112}), right hand ({113-133}), body ({1, 4-11}), and face ({24, 26, 28, 30, 32, 34, 36, 38, 40, 54, 84-91}) to represent each group. Additionally, we select 92, 113, and 54 as the root indices for the hands and face to normalize the keypoints, while the body is not normalized using root coordinates.

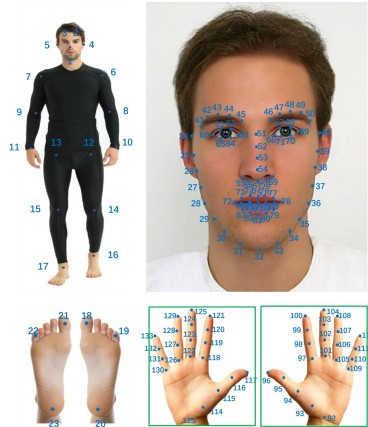

Figure 6: The visualization of the whole-body 133 keypoints, derived from (Jin et al., 2020).

**Feature extraction.** We detail the output dimensions of feature extraction in Table 11. It is important to note that the weights are not shared among each group. Hence, we create three separate linear layers, pose encoders, and temporal encoders to capture the representation of each group individually.

| Layer | Dimensions | Temporal kernel |
|---|---|---|
| Linear | 64 | None |
| Pose encoder | [64, 128, 256] | None |
| Temporal encoder | [256, 256, 256] | 5 |

Table 11: Output dimension of each layer in feature extraction.

**Pre-trained large language model.** We leverage the HuggingFace library for the pre-trained large language model from https://huggingface.co/google/mt5-base.

**Parameters of Uni-Sign.** The parameters of Uni-Sign is shown in Table 12, while the parameters for the compared methods are estimated based on their original papers. Compared to previous methods, Uni-Sign demonstrates significant advantages in both parameter efficiency and performance.

| Method | Visual encoder | Language model | Auxiliary text encoder |
|---|---|---|---|
| FLa-LLM | ResNet18 (11.7M) | MBart-large-cc25 (610.8M) | - |
| Sign2GPT | DinoV2 (ViT-S/14) (22.0M) | XGLM-1.7B (1732.9M) | - |
| SignLLM | ResNet18 (11.7M) | LLaMA-7B (6738.4M) | - |
| $C^2$RL | ResNet18 (11.7M) | MBart-large-cc25 (610.8M) | MBart Encoder (408.2M) |
| Uni-Sign | EfficientNet-B0 + GCN (5.2M + 4.5M) | mT5-Base (582.4M) | - |

Table 12: Comparison of parameters across different methods.

## A.2 Additional Ablation Studies

**Impact of different sub-pose.** To conduct experiments to explore the impact of different sub-pose, we directly use the training from scratch pose-only settings to reduce the time consumption. As presented in Table 13, each sub-pose is indispensable, prompting us to incorporate all sub-poses into our model.

| hands | body | face | MSASL1000 (ISLR) | | CSL-Daily (CSLR) | |
|---|---|---|---|---|---|---|
| | | | P-I Top-1↑ | P-C Top-1↑ | Dev WER↓ | Test WER↓ |
| ✔ | | | 36.07 | 34.20 | 54.3 | 53.9 |
| ✔ | ✔ | | 41.56 | 38.28 | 53.0 | 52.9 |
| ✔ | ✔ | ✔ | **43.45** | **41.10** | **51.2** | **50.7** |

Table 13: Impact of sub-pose in pose-only setting.

## A.3 Pseudocode of Score-aware Sampling Strategy

In order to provide a more detailed explanation of this strategy, we present the pseudocode of score-aware sampling strategy here.

---

**Algorithm 1** Pseudocode of the score-aware sampling strategy in a PyTorch-like style.

---

```
1   # feat_h: pose features of hand, shape [T, 21, C].
2   # score_h: keypoints confidence of hand, shape [T, 21].
3   # coor_h: coordinates of hand, shape [T, 21, 2].
4   # P_samp: sampling probability.
5
6   # Step 1: Pre-define the total duration of the sign language sequence
7   T = feat_h.shape[0]
8
9   # Step 2: Calculate reliability scores (rs) based on keypoint confidence
10  rs = [confidence.mean(-1) for confidence in score_h]
11
12  # Step 3: Calculate sampling scores as 1 - rs
13  sampling_scores = [1 - score for score in rs]
14
15  # Step 4: Perform random sampling
16  sampled_indices = random.choices(range(T), weights=sampling_scores, k=int(T * P_samp))
17
18  # Step 5: Extract RGB frames, pose features and coordinates
19  RGB_frames = [read_hand_image(i) for i in sampled_indices]
20  pose_features = [feat_h[i] for i in sampled_indices]
21  pose_coordinates = [coor_h[i] for i in sampled_indices]
22
23  # Step 6: Interact the RGB modality with the pose modality
24  RGB_features = vision_encoder(RGB_frames)
25  cross_modality_features = PGF(RGB_features, pose_features, pose_coordinates)
26
27  # Step 7: Fuse cross modality features to pose features
28  feat_h[sampled_indices] = cross_modality_features
```

---

## A.4 Qualitative Examples

**Visualization on ISLR and CSLR.** Figure 7 presents representative examples from the ISLR task, showcasing the capability of Uni-Sign to effectively address ISLR challenges. Table 14 presents the CSLR results on the CSL-Daily dataset. Uni-Sign demonstrates powerful SLU capabilities by achieving notable performance on the CSLR task, emphasizing large-scale generative pretraining as a promising direction for scaling up CSLR models. However, failure cases reveal challenges in distinguishing semantically similar words (e.g., " 你 们" (you) → " 你们" (you), " 收" (receive) → " 接受" (receive), " 兴奋" (excited) → " 高兴" (happy)), underscoring the importance of fine-grained control over output targets, which could further enhance model performance and reliability.

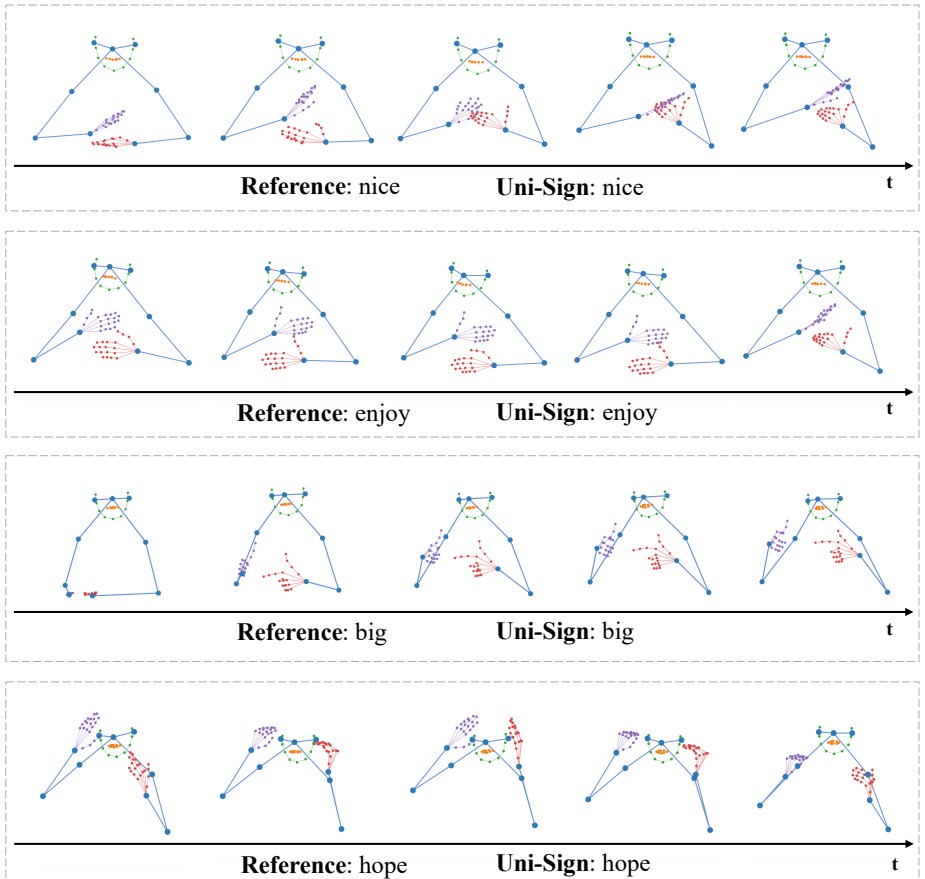

Figure 7: Visualization examples derived from the WLASL and MSASL datasets.

| Reference:
Uni-Sign: | 中午 去 什么 吃 在 学校 饭店
中午 去 什么 吃 在 学校 饭店 |
|---|---|
| Reference:
Uni-Sign: | 人们 排队 排队 清楚 这 是 好 习惯
人们 排队 排队 清楚 这 是 好 习惯 |
| Reference:
Uni-Sign: | 这 项目 是 你们 努力 成功 争取
这 项目 是 你们 努力 成功 |
| Reference:
Uni-Sign: | 哥哥 接受 书 清楚 华 大学 录取 成功 心 兴奋
哥哥 收 书 清楚 华 大学 录取 成功 他 高兴 |

Table 14: Visualization examples derived from the CSL-Daily dataset.

**SLT examples.** In Table 15, 16 and 17, we present several SLT results across different datasets. We found that our model successfully captures semantic information in sign language, generating sentences that are close in meaning to the references, despite occasional differences in sentence structure. However, we also observed that the model sometimes fails to translate complex sentence structures, as demonstrated in the last example of Table 16 and 17.

| | |
|---|---|
| Reference: | 下雪了，今天真冷。 |
| | (It's snowing and it's really cold today.) |
| Uni-Sign: | 下雪了，今天很冷。 |
| | (It's snowing and it's very cold today.) |
| Reference: | 我的爷爷会手语，有很多聋人朋友。 |
| | (My grandfather knew sign language and had many deaf friends.) |
| Uni-Sign: | 我爷爷会打手语，他有很多聋人朋友。 |
| | (My grandfather can sign language and he has many deaf friends.) |
| Reference: | 今天出门忘记带手机，真是太倒霉了。 |
| | (I forgot to bring my mobile phone when I went out today, which is really unlucky.) |
| Uni-Sign: | 今天出门时，我的手机忘了，真倒霉。 |
| | (When I went out today, I forgot my cell phone. What a bad luck.) |
| Reference: | 哥哥接到了清华大学的录取通知书，很高兴。 |
| | (My brother received the admission notice from Tsinghua University and was very happy.) |
| Uni-Sign: | 不难想象，哥哥接到清华大学录取通知书时,心情是多么激动。 |
| | (It is not difficult to imagine how excited my brother was when he received the admission notice from Tsinghua University.) |

Table 15: Translation examples on the CSL-Daily dataset.

| | |
|---|---|
| Reference: | Alright. |
| Uni-Sign: | Okay. |
| Reference: | A little speed. |
| Uni-Sign: | Just a little bit faster. |
| Reference: | A really basic coil to do for ropes that are kind of medium length. |
| Uni-Sign: | The basic coil we're going to do for ropes is some kind of medium length. |
| Reference: | After you are dealt the three cards, you would look down at your cards and you would decide if you wanted to continue to play. |
| Uni-Sign: | I'm going to cut three of these out and I'm going to decide if I want to continue. |

Table 16: Translation examples on the How2Sign dataset.

| | |
|---|---|
| Reference: | An official letter, right. |
| Uni-Sign: | So it's a letter. |
| Reference: | Before the meeting, you should share with your captioner what topics will be discussed, so the captioner will be better prepared for your meeting. |
| Uni-Sign: | Before the meeting, you should discuss with your captioner what topics should be discussed, so that the meeting is smooth. |
| Reference: | After I graduated, I went to Gallaudet University and double majored in Biology and Chemistry. |
| Uni-Sign: | Eventually, I graduated and went to Gallaudet University for my Bachelor s in Biology. |
| Reference: | Besides the dog, Lieutenant Dan, there is a mini horse, pig, llama, hamster, duck and two cats. |
| Uni-Sign: | In addition to the dogs, the dogs and other volunteers include miniature horses, elephants, llamas and hamstrings. |

Table 17: Translation examples on the OpenASL dataset.

### A.5 MORE DISCUSSION ABOUT CSL-NEWS DATASET

To facilitate a more detailed comparison between CSL-News and existing datasets, we provide further analysis of the CSL-News dataset. The vocabulary distribution of CSL-News is presented in Figure 8. In addition to the advantage of a longer duration, we further emphasize several other advantages of CSL-News over existing datasets, as outlined below:

**High diversity of content.** The CSL-News dataset is sourced from news content, encompassing diverse topics such as culture, economy, sports, science and daily life. Compared to small scale datasets (Camgoz et al., 2018; Zhou et al., 2021), it exhibits a more diverse data distribution that is not restricted to specific domains.

**High quality and standardization.** Unlike datasets scraped from YouTube (Li et al., 2020a; Shi et al., 2022; Uthus et al., 2024), CSL-News dataset is derived from news segments featuring sign language experts, ensuring more standardized signing and thereby enhancing the reliability and overall quality of the CSL-News dataset.

Benefiting from the comprehensive CSL knowledge in the CSL-News dataset, models pre-trained on it can acquire robust sign language understanding and generalization capabilities.

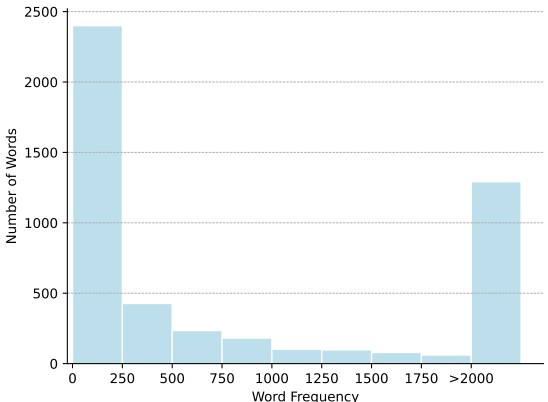

Figure 8: Vocabulary distribution of CSL-News dataset.

## B ETHICS STATEMENT

In this paper, Uni-Sign uses keypoints and cropped hand video clips as input. This ensures that our method not only achieves impressive performance, but also protects the privacy of the Deaf/Hard of Hearing communities.

## C LIMITATIONS

Although Uni-Sign has achieved impressive performance across multiple benchmarks, we still lack a finely annotated, open-domain, large-scale benchmark to further investigate its capabilities and limitations. Furthermore, we fine-tune all parameters of Uni-Sign in all training stages, which presents urgent challenges to computational resources.

## D COMPLETE RESULTS OF EXPERIMENTS

Due to page limitations, some experimental results have been omitted from the main paper. The complete experimental results are provided here to facilitate future research in referencing the relevant results.

| Method | Modality | | MSASL100 | | MSASL200 | | MSASL1000 | | WLASL100 | | WLASL300 | | WLASL2000 | |
| --- | --- | --- | --- | --- | --- | --- | --- | --- | --- | --- | --- | --- | --- | --- |
| | Pose | RGB | P-I | P-C | P-I | P-C | P-I | P-C | P-I | P-C | P-I | P-C | P-I | P-C |
| ST-GCN[†] (Yan et al., 2018) | ✔ | | 50.78 | 51.62 | 44.46 | 45.29 | 34.40 | 32.53 | 50.78 | 51.62 | 44.46 | 45.29 | 34.40 | 32.53 |
| SignBERT (Hu et al., 2021a) | ✔ | | 76.09 | 76.65 | 70.64 | 70.92 | 49.54 | 46.39 | 76.36 | 77.68 | 62.72 | 63.43 | 39.40 | 36.74 |
| BEST (Zhao et al., 2023) | ✔ | | 80.98 | 81.24 | 76.60 | 76.75 | 58.82 | 54.87 | 77.91 | 77.83 | 67.66 | 68.31 | 46.25 | 43.52 |
| SignBERT+ (Hu et al., 2023a) | ✔ | | 84.94 | 85.23 | 78.51 | 79.35 | 62.42 | 60.15 | 79.84 | 80.72 | 73.20 | 73.77 | 48.85 | 46.37 |
| MSLU (Zhou et al., 2024) | ✔ | | 91.54 | 91.75 | 87.79 | 88.58 | 74.07 | 71.81 | 88.76 | 89.25 | 82.04 | 82.71 | 56.29 | 53.29 |
| HMA (Hu et al., 2021b) | | ✔ | 73.45 | 74.59 | 66.30 | 67.47 | 49.16 | 46.27 | - | - | - | - | 37.91 | 35.90 |
| TCK (Li et al., 2020b) | | ✔ | 83.04 | 83.91 | 80.31 | 81.14 | - | - | 77.52 | 77.55 | 68.56 | 68.75 | - | - |
| NLA-SLR (Zuo et al., 2023) | ✔ | ✔ | 90.49 | 91.04 | 88.74 | 89.23 | 72.56 | 69.86 | 91.47 | 92.17 | 86.23 | 86.67 | 61.05 | 58.05 |
| Uni-Sign (Ours) | ✔ | | 93.26 | 93.16 | 90.95 | 91.38 | 77.88 | 76.55 | 92.24 | 92.75 | 88.17 | 88.69 | 63.13 | 60.90 |
| Uni-Sign (Ours) | ✔ | ✔ | 93.79 | 94.02 | 91.02 | 91.56 | 78.16 | 76.97 | 92.25 | 92.67 | 88.47 | 88.92 | 63.52 | 61.32 |

Table 18: ISLR results on various benchmarks. † denotes methods reproduced by (Hu et al., 2021a). Blue and Green denote the best results of previous methods and ours, respectively.

| Method | Modality | | Dev | | | | | Test | | | | |
| --- | --- | --- | --- | --- | --- | --- | --- | --- | --- | --- | --- | --- |
| | Pose | RGB | BLEU1 | BLEU2 | BLEU3 | BLEU4 | ROUGE | BLEU1 | BLEU2 | BLEU3 | BLEU4 | ROUGE |
| *Gloss-based* | | | | | | | | | | | | |
| SLRT[†] (Camgoz et al., 2020) | | ✔ | 37.47 | 24.67 | 16.86 | 11.88 | 37.96 | 37.38 | 24.36 | 16.55 | 11.79 | 36.74 |
| ConSLT (Fu et al., 2023) | | ✔ | - | - | - | 14.80 | 41.46 | - | - | - | 14.53 | 40.98 |
| SignBT (Zhou et al., 2021) | | ✔ | 51.46 | 37.23 | 27.51 | 20.80 | 49.49 | 51.42 | 37.26 | 27.76 | 21.34 | 49.31 |
| SLTUNET (Zhang et al., 2023a) | | ✔ | - | - | - | 23.99 | 53.58 | 54.98 | 41.44 | 31.84 | 25.01 | 54.08 |
| MMTLB (Chen et al., 2022b) | | ✔ | 53.81 | 40.84 | 31.29 | 24.42 | 53.38 | 53.31 | 40.41 | 30.87 | 23.92 | 53.25 |
| CV-SLT (Chen et al., 2024a) | | ✔ | - | - | - | 28.24 | 56.36 | 58.29 | 45.15 | 35.77 | 28.94 | 57.06 |
| TS-SLT (Chen et al., 2022c) | ✔ | ✔ | 55.21 | 42.31 | 32.71 | 25.76 | 55.10 | 55.44 | 42.59 | 32.87 | 25.79 | 55.72 |
| *Gloss-free* | | | | | | | | | | | | |
| MSLU (Zhou et al., 2024) | ✔ | | 33.28 | 21.31 | - | 10.27 | 33.13 | 33.97 | 22.20 | - | 11.42 | 33.80 |
| SLRT[‡] (Camgoz et al., 2020) | | ✔ | 21.03 | 9.97 | 5.96 | 4.04 | 20.51 | 20.00 | 9.11 | 4.93 | 3.03 | 19.67 |
| GASLT (Yin et al., 2023) | | ✔ | - | - | - | - | - | 19.90 | 9.94 | 5.98 | 4.07 | 20.35 |
| NSLT[†] (Camgoz et al., 2018) | | ✔ | 34.22 | 19.72 | 12.24 | 7.96 | 34.28 | 34.16 | 19.57 | 11.84 | 7.56 | 34.54 |
| GFSLT-VLP (Zhou et al., 2023) | | ✔ | 39.20 | 25.02 | 16.35 | 11.07 | 36.70 | 39.37 | 24.93 | 16.26 | 11.00 | 36.44 |
| FLa-LLM (Chen et al., 2024b) | | ✔ | - | - | - | - | - | 37.13 | 25.12 | 18.38 | 14.20 | 37.25 |
| Sign2GPT (Wong et al., 2024) | | ✔ | - | - | - | - | - | 41.75 | 28.73 | 20.60 | 15.40 | 42.36 |
| SignLLM (Gong et al., 2024) | | ✔ | 42.45 | 26.88 | 17.90 | 12.23 | 39.18 | 39.55 | 28.13 | 20.07 | 15.75 | 39.91 |
| C²RL (Chen et al., 2024a) | | ✔ | - | - | - | - | - | 49.32 | 36.28 | 27.54 | 21.61 | 48.21 |
| Uni-Sign (Ours) | ✔ | | 53.24 | 40.54 | 31.65 | 25.27 | 54.34 | 53.86 | 40.96 | 32.02 | 25.61 | 54.92 |
| Uni-Sign (Ours) | ✔ | ✔ | 55.30 | 42.21 | 32.94 | 26.25 | 56.03 | 55.08 | 42.14 | 32.98 | 26.36 | 56.51 |

Table 19: SLT results on CSL-Daily dataset. † and ‡ denote methods reproduced by (Zhou et al., 2021) and (Zhou et al., 2023), respectively. Underline indicates the best gloss-based SLT result.

| Method | Modality | | Dev | | | | | | Test | | | | | |
| --- | --- | --- | --- | --- | --- | --- | --- | --- | --- | --- | --- | --- | --- | --- |
| | Pose | RGB | BLEU1 | BLEU2 | BLEU3 | BLEU4 | ROUGE | BLEURT | BLEU1 | BLEU2 | BLEU3 | BLEU4 | ROUGE | BLEURT |
| *Gloss-free* | | | | | | | | | | | | | | |
| GloFE-VN (Lin et al., 2023) | ✔ | | 21.06 | 12.34 | 8.68 | 6.68 | 21.37 | 36.75 | 21.56 | 12.74 | 9.05 | 7.06 | 21.75 | 36.35 |
| Conv-GRU[†] (Camgoz et al., 2018) | | ✔ | 16.72 | 8.95 | 6.31 | 4.82 | 16.25 | 25.36 | 16.11 | 8.85 | 6.18 | 4.58 | 16.10 | 25.65 |
| I3D-transformer (Shi et al., 2022) | | ✔ | 18.26 | 10.26 | 7.17 | 5.60 | 18.88 | 29.17 | 18.31 | 10.15 | 7.19 | 5.66 | 18.64 | 28.82 |
| OpenASL (Shi et al., 2022) | | ✔ | 20.10 | 11.81 | 8.43 | 6.57 | 20.43 | 31.22 | 20.92 | 12.08 | 8.59 | 6.72 | 21.02 | 31.09 |
| C²RL (Chen et al., 2024a) | | ✔ | - | - | - | - | - | - | 31.46 | 21.85 | 16.58 | 13.21 | 31.36 | - |
| Uni-Sign (Ours) | ✔ | | 49.52 | 36.06 | 28.06 | 22.39 | 42.64 | 60.47 | 49.10 | 35.91 | 28.12 | 22.67 | 42.77 | 60.08 |
| Uni-Sign (Ours) | ✔ | ✔ | 50.84 | 37.82 | 29.83 | 24.16 | 44.58 | 61.28 | 49.35 | 36.32 | 28.55 | 23.14 | 43.22 | 60.40 |

Table 20: SLT results on OpenASL dataset. † denotes methods reproduced by (Shi et al., 2022).

| Method | Modality | | Test | | | | | |
| --- | --- | --- | --- | --- | --- | --- | --- | --- |
| | Pose | RGB | BLEU1 | BLEU2 | BLEU3 | BLEU4 | ROUGE | BLEURT |
| *Gloss-free* | | | | | | | | |
| GloFE-VN (Lin et al., 2023) | ✔ | | 14.9 | 7.3 | 3.9 | 2.2 | 12.6 | 31.7 |
| YouTube-ASL (Uthus et al., 2024) | ✔ | | 37.8 | 24.1 | 16.9 | 12.4 | - | 46.6 |
| MSLU (Zhou et al., 2024) | ✔ | | 20.1 | 7.7 | - | 2.4 | 17.2 | - |
| SLT-IV (Tarrés et al., 2023) | | ✔ | 34.0 | 19.3 | 12.2 | 8.0 | - | - |
| C²RL (Chen et al., 2024a) | | ✔ | 29.1 | 18.6 | 12.9 | 9.4 | 27.0 | - |
| FLa-LLM (Chen et al., 2024b) | | ✔ | 29.8 | 19.0 | 13.3 | 9.7 | 27.8 | - |
| SignMusketeers (Gueuwou et al., 2024) | | ✔ | 41.5 | 27.2 | 19.3 | 14.3 | - | - |
| SSVP-SLT (Rust et al., 2024) | | ✔ | 43.2 | 28.8 | 20.8 | 15.5 | 38.4 | 49.6 |
| Uni-Sign (Ours) | ✔ | | 40.4 | 26.8 | 19.3 | 14.5 | 34.3 | 48.6 |
| Uni-Sign (Ours) | ✔ | ✔ | 40.2 | 27.1 | 19.7 | 14.9 | 36.0 | 49.4 |

Table 21: SLT results on How2Sign dataset.

