# OpenReview forum: "Uni-Sign: Toward Unified Sign Language Understanding at Scale"
_ICLR.cc/2025/Conference — ICLR 2025 Poster_

### Official Review · Reviewer_pQAF · 2024-10-30

**Soundness:** 3
**Presentation:** 2
**Contribution:** 3
**Rating:** 6
**Confidence:** 4

**Summary:**

This paper has two main contributions. 1) A Uni-Sign method for tackling the three sign language understanding tasks in a unified manner. The model first pre-trains on a large sign language dataset via language modeling, then is fine-tuned on each of the individual tasks separately. 2) A CSL-News dataset, which is a large-scale Chinese Sign Language dataset. Some other minor architectural designs are also proposed. Overall, the proposed method performs quite well across the three sign language understanding tasks, and particularly performs well in Sign Language Translation.

**Strengths:**

Developing a unified approach to handle the various sign language understanding tasks is meaningful. In some sense, the work extends some recent LLM-based sign language understanding works by including the aspect of unifying across the sign language understanding tasks.

The authors introduce a new large-scale sign language dataset for Chinese Sign Language. This dataset could be quite useful for further progress in the field.


The experiment results are quite impressive, especially on the gloss-free SLT task. In my opinion, gloss-free SLT is the setting that is the closest to real applications, so this is quite good.

**Weaknesses:**

The proposed method is not very novel. The proposed pre-training approach is to train the model in a language modelling manner, while also using visual features from the sign videos. Then, for the fine-tuning, the language modelling loss is again used for the various tasks. There are some minor contributions, such as a prior-guided fusion module and a score-aware sampling strategy, but these do not seem quite so substantial.

I think that in the related works discussion, there should be a part discussing some other works in other fields employing language modelling (or sequence modeling) for tackling various tasks in a unified manner. For instance, this has been done for image-based tasks, and may have also been done for pose-based tasks. This will give the reader a better understanding of the developments of the “unifying via language modeling” paradigm.


More specific concerns and questions are in the “Questions” section.

**Questions:**

In Table 6, the performance of the proposed method is somewhat lower than the existing baseline SSVP-SLT. Although it is not a very big issue to me, but I would like to know more about it. Why is this the only (rather large) SLT dataset where the proposed method achieves sub-optimal results?

The ablation results shown in Table 7 are rather strange as compared to Tables 8-10, because the settings are different. Table 7 runs experiments on ISLR and CSLR, Tables 8-10 run experiments on CSL-Daily for CSLR and SLT. Why are these different? Moreover, Table 7 and 8 are run in the pose-only setting while Tables 9 and 10 are in the RGB-Pose setting, why should this be the case?
 Furthermore, some of the more important experiments (Table 7 and 8 in my opinion) should be evaluated on all three different sign language understanding tasks.


What is the impact of the pre-training? This crucial aspect as not been evaluated properly. For instance, what if the model is trained only using the fine-tuning stage (Stage 3), but for a longer time (i.e., matching the overall training time of the pre-train then fine-tune approach)? How does this affect the performance? This is important as it shows us the benefits of pre-training. Although some results have been provided in table 7, the results and implications are not clear to me. Furthermore, the task-specific training settings and details have not been mentioned.

---

### Official Review · Reviewer_3Yy5 · 2024-11-02

**Soundness:** 2
**Presentation:** 4
**Contribution:** 4
**Rating:** 6
**Confidence:** 5

**Summary:**

This paper propose Uni-Sign, a unified pre-training framework that eliminates the gap between pre-training and downstream SLU tasks through a large-scale generative pre-training strategy and a novel fine-tuning paradigm. It also introduce CSL-News, a large-scale Chinese Sign Language (CSL) dataset containing1,985 hours of videos paired with textual annotations.

**Strengths:**

Pros:
1. This work proposes a unified framework to conduct pretraining and finetuning, which demostrates novelty.
2. This work shows promising performance across a wide range of benchmarks.
3. The paper is easy to understand.

**Weaknesses:**

Questions and cons:
1. During the data curation process, the authors use a ASR toolkit (FunASR) to convert the speech into texts as labels. There exist some problems. First, as the speech signal has time delay with the sign language expressed by the signer, how to assure that the temporally cropped clips are exactly aligned with the transcribed texts? Second, the authors have stated the averaged length of words are 40 and the averaged length of clips are 9.5s. It's very hard to express 40 words within 9.5s for a signer. Thus, it's most probably that the signer has neglected some meanings in the sentence, and only expressed a part of mearnings in the signs. In this condition, the signs are probably not aligned with the transcribed texts. Third, i observed that in the paper, the authors don't organize a double-check process for the cropped videos from the TV shows to check the alignment between texts and clips, the correctness of transcribed texts,  the correctness of transcribed signs and other things. Thus, how to assure the compleness and correctness of the curated datasets?
2. During the experiments for CSLR, PHOEXNI14 and PHOENIX14-T are also broadly used datasets. Why not report the results on these datasets? It's due to the language gap between pretraining data and downstream data? How about the performance on these two datasets?
3. In table 3 and table 5, some other numbers are bolded except the results reported by the proposed method. The authors may clarify on this or use another way to emphsize the results.

**Questions:**

See above

---

### Official Review · Reviewer_CGke · 2024-11-03

**Soundness:** 3
**Presentation:** 3
**Contribution:** 3
**Rating:** 8
**Confidence:** 3

**Summary:**

This paper presents Uni-Sign, a novel framework for Sign Language Understanding (SLU) that leverages large-scale generative pre-training and a unified fine-tuning paradigm. The paper presents a well-motivated and well-executed approach to SLU. The introduction of the CSL-News dataset and the innovative Uni-Sign framework are significant contributions to the field, demonstrating state-of-the-art performance across various SLU tasks. The paper is well-written and clearly explains the proposed methodology and experimental results.  The authors make several notable contributions:
•Introduction of CSL-News: The authors introduce CSL-News, a large-scale Chinese Sign Language (CSL) dataset comprising 1,985 hours of video-text pairs. This dataset significantly surpasses existing CSL datasets in size and diversity
•Unified Pre-Training and Fine-Tuning:  During fine-tuning, it treats downstream SLU tasks, such as isolated sign language recognition (ISLR), continuous sign language recognition (CSLR), and sign language translation (SLT), as a single SLT task. This unified approach facilitates seamless knowledge transfer and eliminates the need for task-specific fine-tuning methods.
•Prior-Guided Fusion (PGF) Module: To address the limitations of inaccurate keypoints, the authors propose a PGF module that fuses pose and RGB information using keypoint coordinates as priors.
•Score-Aware Sampling Strategy: The authors introduce a score-aware sampling strategy to improve computational efficiency.
•Comprehensive Evaluation: The paper includes a comprehensive evaluation of Uni-Sign across various SLU benchmarks, demonstrating its superior performance in ISLR, CSLR, and SLT tasks.

**Strengths:**

Originality:
1. The paper presents Uni-Sign, a novel unified pre-training framework for Sign Language Understanding (SLU) that bridges the gap between pre-training and downstream tasks by treating them as a single Sign Language Translation (SLT) task during fine-tuning. This approach deviates from previous methods that relied on indirect pretext tasks or were limited by data scale and transfer capability
2. The authors introduce CSL-News, a large-scale Chinese Sign Language (CSL) dataset containing 1,985 hours of video with text annotations, considerably larger than existing CSL datasets. his dataset enables effective large-scale pre-training, addressing a gap in CSL resources compared to American Sign Language (ASL) and British Sign Language (BSL)
3. The paper proposes a Prior-Guided Fusion (PGF) module that utilizes keypoint coordinates as priors to model fine-grained spatial consistency between pose and RGB modalities, going beyond simple spatial-temporal fusion techniques. This approach addresses the representational gap between modalities and leverages keypoints to enhance accuracy.
4. A score-aware sampling strategy is introduced to address the computational challenges of RGB-pose fusion by selectively choosing RGB frames corresponding to low-confidence keypoints, balancing performance with speed

Quality:
1. The paper is well-written and presents a clear and comprehensive methodology. The authors provide detailed descriptions of their approach, including data curation, pre-training and fine-tuning strategies, and multi-modal fusion techniques
2. The ablation studies thoroughly investigate the contribution of each key component, offering insights into the model's performance and the impact of design choices
3. Quantitative results show that Uni-Sign surpasses previous state-of-the-art methods on multiple benchmarks, including significant improvements in BLEU4 scores for SLT tasks

Clarity:
1. The paper is well-organized and easy to follow.
2. Figures and tables effectively illustrate the framework, data distribution, and experimental results
3. Mathematical notations and equations are clearly defined and explained
4. Qualitative translation examples provide further insights into the model's capabilities

Significance:
1. The introduction of the CSL-News dataset addresses a significant need for large-scale CSL resources, potentially fostering advancements in CSL research
2. The unified pre-training and fine-tuning framework with a generative approach demonstrates a promising direction for improving SLU performance, particularly for SLT tasks
3. The proposed PGF module and score-aware sampling strategy offer effective solutions for multi-modal fusion and computational efficiency, potentially benefiting future SLU research
4. The paper's findings have implications for advancing sign language technologies, promoting accessibility and communication for the Deaf/Hard of Hearing community
5. The authors' commitment to open-sourcing the code and dataset further contributes to the significance of the work, facilitating reproducibility and future research in SLU

**Weaknesses:**

1. Discussion on Computational Complexity: While the authors introduce a score-aware sampling strategy to improve efficiency, a more in-depth discussion on the computational complexity of Uni-Sign would be beneficial. This could include analyzing the trade-offs between accuracy and computational cost for different sampling probabilities and exploring potential optimizations.
2. Further Analysis of CSL-News: While the paper describes the creation of CSL-News, further analysis of the dataset's characteristics, such as vocabulary distribution and linguistic complexity, would be valuable. This would provide a more comprehensive understanding of the dataset's potential and limitations.
3. Cross-Dataset Generalization: Evaluating Uni-Sign's performance on unseen sign language datasets would demonstrate its generalization capabilities. This could involve fine-tuning the pre-trained model on a different CSL dataset or even a dataset from another sign language, like American Sign Language (ASL). Successful cross-dataset generalization would highlight the robustness of the learned representations and the effectiveness of the unified approach.
4. Analysis of Error Patterns: A qualitative analysis of the translation errors made by Uni-Sign would provide valuable insights into its limitations and potential areas for improvement. This could involve categorizing errors based on linguistic features, such as sentence complexity, sign ambiguity, or finger-spelling. Identifying common error patterns could guide future research directions.
5. Exploration of Multi-Signer Scenarios: The authors mention their interest in exploring SLU tasks in complex scenarios, such as multi-signer situations. Including preliminary experiments or discussions on adapting Uni-Sign to handle such scenarios would further enhance the paper's impact and contribution to the field.

**Questions:**

The paper in general addressed the ideas and motivations it introduces. The following question will help add more comprehensive understanding.
Generalization and Applicability
1. Multilingual Evaluation: The sources primarily focus on CSL and ASL. Could the authors comment on the applicability of Uni-Sign to other sign languages? How might the model's architecture and pre-training strategies need to be adapted for multilingual SLU? This is important to assess the generalizability of Uni-Sign and its potential impact on a broader range of sign language communities
2. Multi-signer Scenarios: How well does Uni-Sign perform in situations involving multiple signers? What challenges might arise in such scenarios, and how could the model be modified to handle them effectively? Addressing this question would provide a more realistic assessment of Uni-Sign's capabilities in real-world applications where multiple signers may be present

Comparison and Analysis
1. Comparison with LLM-based SLT Methods: Recent studies like Sign2GPT and Sign-LLM have explored the use of LLMs for gloss-free SLT. Could the authors provide a comparative analysis of Uni-Sign against these LLM-based approaches? This would help clarify Uni-Sign's contributions and position it within the broader landscape of SLT research
2. In-depth Analysis of the Unified Fine-tuning Paradigm: How does the shared objective function influence the performance of individual tasks like ISLR and CSLR? Are there any potential task-specific adaptations that could be incorporated within the unified framework to further optimize performance? This analysis would provide a more nuanced understanding of the paradigm's strengths and weaknesses

---

### Official Review · Reviewer_y6Ev · 2024-11-04

**Soundness:** 2
**Presentation:** 3
**Contribution:** 3
**Rating:** 6
**Confidence:** 2

**Summary:**

This paper proposes Uni-Sign, a unified pre-training framework for sign language understanding (SLU) tasks, addressing the challenges in existing methods that struggle with transferring knowledge across different tasks. The framework uses a new large-scale dataset, CSL-News, which contains 1,985 hours of Chinese Sign Language (CSL) videos paired with textual annotations. Extensive experiments demonstrate that Uni-Sign achieves state-of-the-art performance across multiple SLU benchmarks.

**Strengths:**

1. Uni-Sign effectively unifies multiple SLU tasks, such as isolated sign language recognition (ISLR), continuous sign language recognition (CSLR), and sign language translation (SLT), under a single framework.
2. The introduction of CSL-News, a substantial CSL dataset, provides a significant resource for the SLU field and addresses the limitations of prior smaller datasets.

**Weaknesses:**

1. Compared to other datasets, what unique advantages or characteristics does the proposed CSL-News dataset offer besides its longer duration? Additionally, why is the vocabulary size relatively limited, and could the restricted language variety impact pre-training effectiveness?
2. In the comparisons of downstream tasks in Section 4.3, did other methods also use the CSL-News dataset for pre-training? If not, does this raise any concerns about fairness in the comparisons?
3. In the comparative experiments, while high-performing results are analyzed, the reasons behind lower performance should also be provided, such as in Tables 4 and 6.
4. In Tables 3 to 6, what would the results of Uni-Sign be if it used only RGB video?
5. How do the computational costs, inference time, and memory usage of the proposed model compare to other methods? Does Uni-Sign maintain a competitive advantage in these aspects?
6. The manuscript includes numerous comparative results, but it lacks visualizations to intuitively demonstrate the model’s effectiveness. More visual presentations for each downstream task are recommended.

**Questions:**

Please refer to the Weakness section above. If the authors can address these concerns, I would consider raising the rating.

---

### Official Review · Reviewer_w7tC · 2024-11-05

**Soundness:** 2
**Presentation:** 3
**Contribution:** 2
**Rating:** 6
**Confidence:** 4

**Summary:**

This paper presents a new pre-training framework that bridges the gap between pre-training and downstream sign language understanding tasks through a large-scale generative pre-training strategy and a novel fine-tuning paradigm that achieves impressive performance in multiple benchmark tests.

**Strengths:**

1. The Uni-Sign framework proposed by the authors utilizes a large-scale generative pre-training strategy and a novel fine-tuning paradigm to bridge the gap between pre-training and downstream sign language understanding tasks in traditional approaches.

2. The Uni-Sign framework achieves significant performance gains on both sign language recognition and translation tasks, and experiments are conducted on multiple datasets.

3. The related work of paper is adequate, investigating research on sign language tasks including pre-training strategies, dataset development, and so on, from a variety of perspectives.

**Weaknesses:**

1. The paper is not clear and detailed enough to explain the score-aware sampling strategy, and does not give a detailed analysis of the process or a corresponding explanation in Figure 5, which could lead to potential misunderstandings or errors.

2. The author omitted experimental results on several widely used datasets, such as Phoenix14, Phoenix14T, USTC-SLR 500, USTC-CSL100, etc.

3. As shown in Tables 4 and 6, the proposed Uni-Sign method does not achieve the best performance on multiple datasets of continuous sign language recognition and sign language translation. It even performs worse when more modalities are introduced, which makes me worried about the performance of this work.

4. The number of parameters of the model is not mentioned in the paper. This feedback highlights the importance of including these key performance metrics, as they are critical for evaluating the practicality of the model.

5. It is recommended that the authors make font color changes for the tables throughout the article, due to the large amount of experimental data, while bolding may mislead the reader, especially for Tables 3 through 6.

**Questions:**

1. Although there is a difference for traditional sign language recognition methods employing such means as MLP and CTC loss, the authors propose for different tasks still use different supervision, for example, words, glosses, and sentences, and why it is still referred to as a unified paradigm.

2. In Fig. 5, why the feature information of the face has to be forwarded to the left Pose Encoder after it has been encoded by the Pose Encoder is not mentioned in the paper..

3. In line 479 of the paper, the authors show a boost of 1.36 on BLEU-4, but the corresponding value is not found in Table 9.

---

### Meta-Review · Area_Chair_z6oR · 2024-12-16

**Metareview:**

This paper proposes a unified pre-training framework for Sign Language Understanding (SLU) that bridges the gap between pre-training and downstream tasks, including isolated sign language recognition (ISLR), continuous sign language recognition (CSLR), and sign language translation (SLT). This is achieved through a large-scale generative pre-training strategy and a unified fine-tuning paradigm. The key strengths of the paper include its innovative unified framework that addresses knowledge transfer challenges in SLU, the introduction of CSL-News, which is significantly larger and more diverse than existing datasets (1,985 hours of Chinese Sign Language videos paired with textual annotations), and the inclusion of PGF and score-aware sampling, which improve multi-modal learning efficiency and accuracy. The comprehensive evaluation on various SLU benchmarks further demonstrates the robustness and effectiveness of the proposed approach. All reviewers provided positive feedback, with recommendations of "accept" or "marginally above the acceptance threshold."

**Additional Comments On Reviewer Discussion:**

Several key concerns were raised by the reviewers, including: 1) a lack of sufficient clarity and detail in explaining the score-aware sampling strategy, 2) the absence of a specification for the number of parameters in the model, 3) the results of using only RGB video, 4) a lack of discussion on computational complexity, 5) the absence of an analysis of error patterns, and 6) uncertainty regarding how the temporally cropped clips are aligned exactly with the transcribed texts. In the rebuttal, the authors comprehensively addressed each of these points and effectively responded to all major comments. As a result, all reviewers are now positive about the paper.

---

### Decision · Program_Chairs · 2025-01-22

Accept (Poster)